# Exposures in the Indoor Environment and Prevalence of Allergic Conditions in the United States of America

**DOI:** 10.3390/ijerph18094945

**Published:** 2021-05-06

**Authors:** Janvier Gasana, Boubakari Ibrahimou, Ahmed N. Albatineh, Mustafa Al-Zoughool, Dina Zein

**Affiliations:** 1Department of Environmental and Occupational Health, Faculty of Public Health, Kuwait University, Hawally P.O. Box 24923, Safat 13119, Kuwait; mustafa.alzoughool@ku.edu.kw (M.A.-Z.); dina.zein@ku.edu.kw (D.Z.); 2Department of Biostatistics, Stempel School of Public Health, Florida International University, Miami, FL 33199, USA; bibrahim@fiu.edu; 3Department of Community Medicine and Behavioral Sciences, Faculty of Medicine, Kuwait University, Jabriya P.O. Box 46300, Safat 13119, Kuwait; ahmed.albatineh@ku.edu.kw

**Keywords:** allergy, asthma, wheeze, rhinitis, home allergens, household dust, mildew, cockroach, dust mites, pets, endotoxin, population study, prevalence study, NHANES

## Abstract

Our study examines the association of the presence of mildew, cockroaches, and pets in homes as well as household dust allergens with the prevalence and/or severity of allergic diseases. No study has concurrently assessed home environment exposures in relation to allergic conditions in the general US population. Data from 5409 participants from the 2005–2006 National Health and Nutrition Examination Survey (NHANES) living in their current homes for ≥one year were analyzed. Multivariate logistic regression analyses between home exposures and allergic diseases prevalence and severity were performed. In adjusted analyses, mildew was associated with higher current asthma, allergies, and allergic rhinitis prevalence; endotoxin, with higher current asthma prevalence; and dust *Canis familiaris* (Can f) 1, with higher allergic rhinitis prevalence. However, presence of cockroaches and dust *Dermatophagoides farinae* (Der f) 1 were associated, respectively, with lower current asthma and allergies prevalence. Presence of mildew, dust Der f1, *Dermatophagoides pteronyssinus* (Der p) 1, *Felis domesticus* (Fel d) 1, and endotoxin were all associated with asthma and/or wheeze severity. Non-atopic asthma was more frequent with mildew and/or musty smell dust and higher dust Fel d1 concentration, while atopic asthma was more prevalent with higher Can f1 and endotoxin concentrations in dust. This study confirms previous relationships and reports novel associations, generating hypotheses for future research.

## 1. Introduction

Allergic diseases designate a group of conditions (rhinitis, conjunctivitis, asthma, and atopic dermatitis) occurring in predisposed individuals and are among the most common illnesses in the world [1]. Their prevalence has increased in both developed and developing countries over the past decades, with now up to 40% of the global population having at least one allergic condition [2]. Asthma alone affects about 235 million people worldwide, including 25.9 million people in the US, where its annual economic cost is estimated to be more than $80 billion [3]. Asthma is the most prevalent chronic disease of childhood in the US, and the third cause of hospitalization among children under the age of fifteen [4]. The increasing trends in asthma prevalence has been linked to changing environmental exposures, particularly in developing countries [5]. Emissions of vehicular pollution in industrialized countries and industrial pollution in countries undergoing industrialization contributes to a large extent of deteriorating air quality [6]. These pollutants contribute to substantial burden of chronic disease and related acute exacerbations in many European urban areas [7]. Climate change is also projected to contribute to the increase in asthma prevalence and exacerbations, mainly through increase in ground level ozone and particulate matter pollution, and increased exposure to pollens resulting from shifts in flowering time and pollen initiation [8].

Indoor allergens are known to contribute to asthma exacerbations; however, their role as risk factors for asthma and other allergic diseases is less clear [9]. Moreover, most studies have addressed this question by evaluating associations between some indoor allergens and specific allergic diseases on limited samples, failing to account for other certainly relevant environmental exposures despite the tendency of allergens to cluster together in high levels [10]. Therefore, in this report, we concurrently evaluated the association of multiple household exposures with the prevalence and severity of allergic conditions in a sample representative of the US population, adjusting for other potential confounding allergens or environmental exposures among other covariates.

## 2. Materials and Methods

### 2.1. Participants and Data Source

We used data from the National Health and Nutrition Examination Survey (NHANES) conducted from 2005 to 2006 by the National Center for Health Statistics (NCHS) of the Centers for Disease Control and Prevention (CDC) [11]. The NHANES is an ongoing cross-sectional survey of the US non-institutionalized civilian population selected using a complex multistage sampling design to derive a representative sample of the US population.

For our study, a total of 6957 NHANES participants had data on house dust allergens; among them, 6948 were interviewed on allergic diseases. After exclusion of 1418 participants who lived in their current home for less than a year and exclusion of 121 participants with missing data, the final sample included 5409 subjects. NHANES protocols were approved by the institutional review boards of the NCHS (NHANES 2005–2006 NCHS IRB: Protocol #2005-06) and informed consent was obtained from all participants.

The NHANES comprises both a survey and an examination. The survey includes questions on demographics, socioeconomic status (SES), diet, tobacco use, medical conditions and diseases, and other health-related topics. The examination is done in specially designed and equipped trailers, staffed by a team that comprises a physician, medical and health technicians, and dietary and health interviewers. Bilingual (Spanish–English) interviewers are provided. Health measurements taken by this mobile team include weight, height, blood pressure, and muscle strength. In addition, blood samples are drawn, and urine samples are collected. Details on study design and procedures can be found in the NHANES website (http://www.cdc.gov/nchs/nhanes.htm) [11].

### 2.2. Procedures: Data Collection

#### 2.2.1. Environmental Exposures

Environmental exposures were assessed using questionnaires for the presence of mildew (“In the past 12 months, has your home had mildew odor or musty smell?”), cockroaches (“In the past 12 months, have you seen any cockroaches in your home?”), and pets in the home (In the past 12 months, did any dogs, cats or other furry animals live or spend time inside your home?”).

#### 2.2.2. House Dust

A Sanitaire™ Model 3683 vacuum cleaner (Bissell Inc.: Grand Rapids, MI, USA) and a Mitest™ Dust Collector (Indoor Biotechnologies, Inc., Charlottesville, VA, USA) was used to collect combined bed and bedroom dust samples from each participant’s house. A 1-square yard surface on both bed and adjacent floor was independently vacuumed for two minutes. Dust samples were analyzed for a panel of allergens (Cockroach: Bla g 2; dog: Can f 1; cat: Fel d 1; dust mites: Der p 1 and Der f 1; mouse: Mus m 1; and rat: Rat n 1) using the Indoor Biotechnologies ELISA MARIA^®^ Multiplex Array assay (Indoor Biotechnologies, Inc., Charlottesville, VA, USA). The Bla g1 assay was performed at Air Quality Sciences, Inc. using enzyme-linked immunosorbent assay (ELISA) test kit from Indoor Biotechnologies Inc. The *Aspergillus fumigatus* assay was performed at Air Quality Sciences, Inc. using a custom-prepared ELISA test from Greer Laboratories (Lenoir, NC, USA). Dust endotoxin assay was performed at the University of Iowa laboratory using a Limulus amebocyte lysate assay based on the sensitivity of an enzymatic clotting cascade in the amebocytes found in the hemolymph of the horse-shoe crab *Limulus polyphemus*.

#### 2.2.3. Immunoglobulin E (IgE) Levels

Specific IgE levels were measured against 15 aeroallergens (*Alternaria alternata*, *Aspergillus fumigatus*, Bermuda grass, birch, cat dander, cockroach, dog dander, dust mites, mouse urine proteins, oak, ragweed, rat urine proteins, Russian thistle, rye grass) using the Pharmacia Diagnostics ImmunoCAP 1000 System (Kalamazoo, MI, USA).

#### 2.2.4. Laboratory Tests

Laboratory tests included, among other things, the drawing of blood and analysis of it for exposure to tobacco smoke, environmental pollutants, and dust allergens. Details on the laboratory results may be found on NHANES website (http://www.cdc.gov/nchs/nhanes.htm) [11].

#### 2.2.5. Assessment of Atopic Diseases

Asthma and other atopic diseases were assessed using a questionnaire, which does not ask for medications. Current asthma was defined as an affirmative answer to both of the following questions: “*Has a doctor or other health professional ever told you that you/Sample Person (SP) had/have/has asthma?*” and “*Do you/Does SP still have asthma?*” Wheezing was classified as positive when the response was yes to the following question: “*In the past 12 months, have you/has SP had any wheezing or whistling in chest?*” Allergies were classified as positive based on the questions: “*Has a doctor or other health professional ever told you that you/SP had/have/has allergies?*” and “*During the past 12 months, have you/has SP had any allergy symptoms or allergy attack?*” Allergic rhinitis was classified as positive when the participants said yes to the following question: “*During the past 12 months, have you/has SP had a problem with sneezing, or a runny, or blocked nose when you did not have a cold or the flu?*”, and presented specific IgE ≥ 0.35 kU/L against any aeroallergen.

Asthma and wheezing severity were assessed by the number of emergency room visit(s) for asthma (“*During the past 12 months, have you/has SP had to visit an emergency room or urgent care center because of asthma?*”) or acute medical care for wheezing (“*In the past 12 months, how many time have you/has SP gone to the doctor’s office or hospital emergency room for one or more of these attacks of wheezing or whistling*”). Medical care for wheezing was categorized into 1, 2 or 3+ doctor’s office/emergency room (ER) visits. The NHANES did not collect data on pharmacotherapy for respondents with asthma.

Atopic asthma and wheezing were defined as current asthma or wheezing with specific IgE ≥ 0.35 kU/L, while non-atopic asthma or wheezing was defined as current asthma or wheezing with specific IgE < 0.35 kU/L.

#### 2.2.6. Covariates

Data on age, gender, race/ethnicity, family income, parental asthma, and presence of a smoker in the household were collected using questionnaires. Participants were also asked about pet removal or avoidance because of allergies. Estimation of family income to poverty ratio was conducted using guidelines and adjustment for family size, year and state. Participants’ weight and height were measured, and body mass index (BMI) was calculated as weight in kilograms divided by height in meters squared. BMI was divided into underweight (<18.5 kg/m^2^), normal weight (18.5–24.9 kg/m^2^), and overweight (≥25 kg/m^2^).

### 2.3. Statistical Analysis

*p*-values for differences in proportions or means by outcome status were calculated using chi-square test for categorical variables and using Student *t*-test for continuous variables. After normal distribution testing using the Kolmogorov–Smirnov test, dust allergen concentrations were log-transformed and geometric means were calculated for descriptive analysis from log-transformed estimates. Parametric univariate and multivariate models with conditional logistic regression analyses were used to determine the association between different variables and the risk of the three allergic conditions. Odds ratios (ORs) and 95% confidence intervals (CIs) were calculated for the different variables. In multivariate logistic regression analysis, presence of mildew, cockroaches, and pets in homes was dichotomized, while dust allergen levels were divided into quartiles comparing the highest to the lowest quartile. We used dichotomization because a high percentage of exposure levels were under detection limit (See Table A1 Quartile in the Appendix A). The models were adjusted for age, gender, race/ethnicity, family income to poverty ratio, environmental tobacco smoke, BMI, pet avoidance or removal, and potential confounding environmental exposures (associated with the exposure of interest and the outcome). All analyses were performed in STATA (Version 11, STATA Corporation, College Station, TX, USA). In all statistical procedures, NHANES sampling weights and STATA survey commands, taking into account the multistage and complex survey design, were used so that estimates were nationally representative. Because of sampling weights, traditional procedures for goodness of fit could not be performed; instead Stata, syntax svylogitgof, an F-adjusted mean residual test, was used [12]. *p*-values < 0.05 were considered statistically significant.

## 3. Results

The study sample consisted of 5409 participants with a mean age (±standard error—SE of 37.51—±0.32 years—range 6–85 years); 51% of the sample were females and 68.9% were non-Hispanic whites. About 18.5% of participants had a smoker in the household. The weighted prevalence of current allergic diseases was 9.0% for asthma, 16.4% for wheezing, 22.0% for allergy, and 16.1% for allergic rhinitis. Participants with current asthma tended to be residents of households with mildew and/or musty smell or high endotoxin levels, while people without current asthma more commonly reported cockroaches in their houses (in Appendix A Table A1).

Participants with wheezing in the past year tended to live in houses with a dog, a cat, or higher levels of Can f1, Fel d1, or endotoxin, but with lower level of Bla g1 than those without wheezing. People with allergies in the past 12 months tended to live in households with mildew and/or musty smell, a cat, or higher levels of Can f1 or Fel d 1, but with lower levels of Bla g1, Der f1, Der p1, or Rat n1 than those without the symptoms. Participants with allergic rhinitis in the past year tended to live in houses with mildew and musty smell, a dog, or higher levels of Can f1, but with no report of seen cockroaches in their homes and lower levels of Bla g1, Der f1, Der p1, and Mus m1. It shows little correlation between exposures. Similar highest correlation values of 0.25 were observed between Bla g 1 and Bla g 2 and between Can f 1 and Bla g 2.

In multivariate logistic regression (Table 1), mildew and/or musty smell were associated with higher prevalence of current asthma, allergies, and allergic rhinitis. Can f1 was associated with higher prevalence of allergic rhinitis; and dust endotoxin with higher prevalence of current asthma. Conversely, the presence of cockroaches was associated with lower prevalence of current asthma and Der f1, with lower prevalence of allergies in the past 12 months.

When assessing asthma severity (Table 2), Der f1 was positively associated with one ER/doctor visit for wheeze; Der p1 with two visits; and mildew and/or musty smell, Fel d1, and endotoxin with three visits in past 12 months.

In Table 3, mildew and/or musty smell in house and Fel d1 were associated with higher prevalence of non-atopic asthma. Can f1 was associated with higher prevalence of atopic asthma and wheeze. Endotoxin was associated with higher prevalence of atopic asthma. Negative associations were found between Der f1 and non-atopic asthma, as well as between the presence of cockroaches and atopic asthma.

## 4. Discussion

This study examined the association between a wide range of household environmental exposures and atopic diseases in a representative sample of the US population. Mildew was associated with higher prevalence of asthma, allergies, and allergic rhinitis. Can f1 was associated with higher prevalence of allergic rhinitis, and endotoxin was associated with higher prevalence of asthma. However, presence of cockroaches was associated with lower asthma prevalence and Der f1 with lower prevalence of allergies. Among people with wheeze, mildew, and/or musty smell, Der f1, Der p1, Fel d 1, and endotoxin were associated with higher symptom severity. By atopic status, mildew and/or musty smell dust and Fel d1 were associated with higher prevalence of non-atopic asthma, Can f1 with higher prevalence of atopic asthma and wheeze, endotoxin with higher prevalence of atopic asthma.

Previous national or multicenter studies investigated the associations of some dust allergens with asthma or atopic diseases either in children [13] or adults [14]. A report has also looked at the burden of several allergens in US housing units and assessed the link between number of allergens and asthma among the occupants [10]. Salo et al. evaluated the relationship between sensitization to 19 aeroallergens and food items and allergic conditions using NHANES 2005–2006 data [15]. About the association exposure and sensitization, atopic sensitization is defined as immunoglobulin E (IgE) positivity or prick test positivity to any common food or air born allergens. Atopic sensitization is an important risk factor for the development of asthma and allergic diseases. Many complicated factors, such as genetics, age, exposure time, exposure amount to allergens and environmental factors among other affect sensitization to allergens [16]. Some studies found a significant association between household dust mite and high specific IgE to dust mites, while other studies did not find a relationship between pet allergens in house dust and sensitivity to the same pet allergens [17,18,19]. To our knowledge, this is the first study to simultaneously assess several household exposures and the most common atopic conditions as well as their severity in a sample representative of the general US population.

In line with our findings, the association between presence of mildew in homes and allergic diseases has been previously identified, [20,21,22]. Other related factors for which causal relationship with asthma exacerbations has been established include exposures to indoor dampness and dampness-related agents both in children), and in adults. [23]. Similarly, exposure to environmental tobacco smoke (ETS) among pre-school children and exposure to endotoxin were associated with exacerbation of asthma [23]. Exposure to fungal spores has been linked to increased severity of asthma and asthma symptoms while fungal antigens have been associated with reduction in asthma symptoms [24]. On the other hand, there was limited evidence of association between asthma exacerbations and exposures to dust mite, cockroach, dog, fungi, and dampness-related agents [23]. A positive bronchial provocation tests with dog allergens have been correlated with sensitization to dog allergen [24]. The effects of exposure to dog allergen have been found among children sensitized to dogs; however, the evidence of this sensitization among non-sensitized adults is limited [24]. Allergies to a dog is a widespread problem that affects a large proportion of adults, and in children and adults suffering from asthma, a dog allergy acts as a triggering factor. Furthermore, it has been estimated in Sweden that up to 34%, 33% and 21%, of children with a physician-diagnosed asthma, rhinitis, or eczema, respectively, have dog-sensitization as confirmed by skin prick test [25].

Can f1 is a lipocalin protein from dogs’ salivary glands; although having a dog drastically increases its levels, detectable amounts can be found in dust from houses without dogs [25]. Consistent with our result of a positive association between Can f1 and allergic rhinitis, the Tucson Children’s Respiratory Study concluded that the existence of dogs in a household early in life increased the risk of allergic rhinitis by age six years [26]. Data on dog exposure and allergic rhinitis is limited with conflicting findings: In a meta-analysis including 22,000 children from 11 prospective European birth cohorts, dog ownership was not associated with allergic rhinitis [27]; other studies have even reported a protective effect [28,29] These discrepancies in the literature may be due to lifestyle changes related to atopic diseases (e.g., “healthy pet-keeping effect”) or by the complex mixture of allergens and microbes that animals carry and spread indoors [27,29] The same reasons may explain contradictory literature about the association between cat exposure and atopy. We found a positive relationship between Fel d1 and current asthma prevalence or allergies in previous 12 months, while other studies have found no association [29,30] or a negative association [28] with asthma. Remarkably, among asthmatics, Der f1, Der p1, Fel d1, and endotoxin were highly associated with ER or urgent care visits for asthma. Prior studies on exposure and/or sensitization to dust mites have also reported conflicting results [30,31,32,33,34]. Nelson et al. reported a relationship between sensitivity to Der p and emergency treatment for asthma [35]. Similarly, endotoxin exposure and sensitization to cats have been reported to be positively associated with severity of asthma [22,36,37].

Timing and length of exposure seem to be essential factors influencing the direction of association between some exposures and allergic diseases. As per “hygiene” hypothesis, early life exposure to microbial agents with immune-modulatory properties could up-regulate the maturation of Th1 cells and down-regulate atopic Th2-oriented immunity [38]. It is possible; however, that some of the previous protective relationships were explained by unmeasured confounders. To date, dust mites, including Der p1 and Der f1, are widely recognized as risk factors for allergic diseases. Their cysteine protease activity has been reported to expose tight junctions in the epithelium and to slice lymphocyte surface receptors while their fecal particles have constituents that include ligands for TLR-2 (chitin), TLR-4 (endotoxin), and TLR-9 (mite or bacterial DNA) which could heighten immune reactions favoring IgE antibody reactions [39]. Yet, Von Mutius suggested that the strong association between asthma and dust mite sensitization (widely used as indicator of exposure) could be explained by an individual susceptibility, rather than higher exposure [40]. To support that idea, the author noted the contrast between the low prevalence of house dust exposure among habitants of Las Alamos, New Mexico, and the asthma prevalence (17%) in the community. Intriguingly, negative association between Der f 1 and asthma or wheeze (marginal significance) were seen only in non-atopic participants, but not in atopic ones. In another study, some bacterial taxa were found positively associated with cockroach allergens, and the authors suggested that protection against the development of atopy in earlier life stages can be explained by injection with the microbiome species upon exposure to some dust containing some bacteria [32]. We cannot rule out; however, the possibility that subjects with asthma or allergies may be more likely to take measures to reduce cockroach or dust mite load in their homes, although we adjusted for pet removal and/or avoidance.

We tested for correlation between exposures (see Table A2 in Appendix A) but the correlation matrix showed little correlation between exposures. Similar highest correlations of 0–25 were observed between Bla g 1 and Bla g 2 and between Can f1 and Bla g 2.

### Strength and Limitations

Given the cross-sectional design of our study, causality association between dust allergens and allergic diseases cannot be established. However, we only considered allergic conditions that occurred within the past 12 months and only included participants who lived at their current home for more than a year. Although dust was sampled only once, allergens derived from dust found on the mattress and in the bedroom have been shown to reflect one-year long exposures [41]. Another limitation of our study is that we did not adjust for covariates, such as familial atopy, because it was available for adults 20 years and older only. But, when analyzing for this age group alone, adjusting for familial asthma resulted in the same results. There is also a potential recall bias as in any survey. An additional limitation is the lack of objective assessment since only questionnaires were used in NHANES. NHANES did not make the distinction between houses with carpets and those without carpets, which is also a limitation of our study. In fact carpets concentrate more dust than non-carpeted floors; thus, for example, Heinrich et al. (2003) [41] have standardized approaches for vacuuming carpeted (dusting 1 m^2^) and non-carpeted floors (dusting the whole room). NHANES did not test for Alternaria or Cladosporium allergens, or black molds (e.g., *Stachybotrys chartarum*), which are increasingly acknowledged allergenic sources and may be associated with worse asthma control in some cases (Segura-Medina et al., 2019) [42], this being another limitation of our study. A cluster analysis that Mendy et al. (2020) [43] used would have assessed real life exposures. Indeed, the authors found out that the clustering of endotoxin with allergens in dust from homes with a pet or of people with low SES is associated with asthma and wheeze. A major strength of our study is the broad range of environmental exposures and allergic outcomes examined in a large representative sample of the US population. An additional strength is that we analyzed the data adjusting for dust allergen levels that may have been confounders of each association in previous studies.

## 5. Conclusions

Our study showed that the presence of mildew, Can f1, and endotoxin were associated with higher prevalence of allergic diseases. Mildew and/or musty smell, Der f1, Der p1, Fel d 1, and endotoxin were associated higher asthma or wheeze severity. Non-atopic asthma was more frequent with mildew and/or musty smell dust and higher dust Fel d1 concentration, while atopic asthma was more prevalent with higher Can f1 and endotoxin concentrations in dust. This study confirms previous relationships and reports novel associations, generating hypotheses for future research. Future research should include prospective studies accounting for potentially confounding effects between allergens and allergic diseases.

## Figures and Tables

**Table 1 ijerph-18-04945-t001:** Odds ratios (OR) and 95% confidence intervals (CI) for associations between dust components and allergic conditions, NHANES 2005–2006 (*N* = 5409).

Dust Allergens	Current Asthma		Wheeze in Past 12 Months		Allergies in Past 12 Months		Allergic Rhinitis in Past 12 Months	
	OR (95% CI)	*p*	OR (95% CI)	*p*	OR (95% CI)	*p*	OR (95% CI)	*P*
Mildew or musty smell	1.67 (1.10, 2.53)	0.02	1.19 (0.84, 1.68)	0.31	1.68 (1.29, 2.20)	0.00	1.55 (1.16, 2.06)	0.00
Cockroaches in house	0.51 (0.40, 0.65)	<0.001	0.87 (0.57, 1.32)	0.49	1.03 (0.74, 1.43)	0.84	0.76 (0.48, 1.22)	0.23
Dog in house	0.76 (0.47, 1.24)	0.25	0.92 (0.72, 1.18)	0.49	0.82 (0.51, 1.34)	0.41	1.04 (0.67, 1.61)	0.86
Cat in house	1.08 (0.80, 1.47)	0.58	1.15 (1.00, 1.33)	0.05	1.12 (0.93, 1.35)	0.23	1.08 (0.90, 1.29)	0.40
Furry pet in house	1.00 (0.81, 1.24)	0.98	1.02 (0.81, 1.28)	0.86	1.11 (1.00, 1.24)	0.05	1.18 (0.99, 1.41)	0.07
*Aspergillus fumigatus*	0.85 (0.54, 1.34)	0.46	1.03 (0.73, 1.21)	0.87	0.99 (0.70, 1.42)	0.98	1.32 (0.74, 1.73)	0.54
Bla g 1	1.21 (0.82, 1.77)	0.31	1.08 (0.81, 1.44)	0.57	1.04 (0.83, 1.29)	0.75	1.26 (0.91, 1.76)	0.15
Bla g 2	0.75 (0.47, 1.19)	0.20	0.95 (0.74, 1.21)	0.64	0.99 (0.69, 1.43)	0.97	0.91 (0.73, 1.14)	0.40
Can f 1	1.90 (0.93, 3.86)	0.07	1.43 (0.95, 2.47)	0.08	1.31 (0.55, 3.11)	0.52	1.67 (1.07, 2.63)	0.02
Der f 1	0.63 (0.38, 1.05)	0.07	0.70 (0.48, 1.04)	0.08	0.72 (0.55, 0.95)	0.02	0.76 (0.57, 1.01)	0.05
Der p 1	1.36 (0.81, 2.28)	0.23	1.21 (0.85, 1.73)	0.27	0.91 (0.63, 1.31)	0.59	0.89 (0.62, 1.29)	0.53
Fel d 1	1.37 (0.80, 2.35)	0.23	1.30 (0.64, 2.64)	0.44	1.55 (0.92, 2.61)	0.10	0.94 (0.57, 1.56)	0.80
Mus m 1	0.88 (0.52, 1.51)	0.63	0.91 (0.53, 1.59)	0.73	0.87 (0.64, 1.20)	0.38	0.87 (0.58, 1.31)	0.48
Rat n 1	1.35 (0.91, 1.99)	0.13	1.07 (0.77, 1.50)	0.66	0.96 (0.68, 1.35)	0.80	1.00 (0.77, 1.29)	0.97
Endotoxin	1.44 (1.05, 1.99)	0.02	1.39 (0.94, 2.06)	0.10	1.15 (0.77, 1.71)	0.48	1.06 (0.66, 1.71)	0.79

Abbreviations: OR, odds ratio; CI, confidence interval; Bla g, *Blattella germanica*; Can f, *Canis familiaris*; Der f, *Dermatophagoides farinae*; Der p, *Dermatophagoides pteronyssinus*; Fel d, *Felis domesticus*; Mus m, mouse urinary protein; Rat n, *rat urinary protein*. Odds ratios (OR) calculated using logistic regression analyses adjusting for age, gender, race/ethnicity, family income to poverty ratio, environmental tobacco smoke, BMI, pets’ avoidance or removal, and potential confounding dust allergens. OR estimated for an increase in exposure from lowest to highest quartile of dust allergens concentrations (*Aspergillus fumigatus*, Bla g1 and 2, Can f1, Fel d1, Der p1, Der f1, Mus m1, Rat n1, endotoxin). Significant results are underlined.

**Table 2 ijerph-18-04945-t002:** Odds ratios (OR) and 95% confidence intervals (CI) for associations of dust components with asthma and wheeze severity, NHANES 2005–2006.

Dust Allergens	ER Visit for Asthma		Number of Doctor’s Office or ER Visits for Wheezing					
			1		2		≥3	
	OR (95% CI)	*p*	OR (95% CI)	*p*	OR (95% CI)	*p*	OR (95% CI)	*p*
Mildew or musty smell	0.48 (0.21, 1.11)	0.10	1.12 (0.55, 2.27)	0.73	0.72 (0.40, 1.29)	0.25	2.14 (1.01, 4.51)	0.04
Cockroaches in house	1.02 (0.20, 5.31)	0.98	0.81 (0.35, 1.87)	0.60	0.97 (0.36, 2.66)	0.95	0.82 (0.37, 1.81)	0.60
Dog in house	1.42 (0.35, 5.78)	0.60	1.09 (0.46, 2.59)	0.83	1.35 (0.63, 2.92)	0.42	2.55 (0.93, 6.97)	0.07
Cat in house	1.17 (0.53, 2.57)	0.68	0.86 (0.47, 1.56)	0.60	1.25 (0.90, 1.73)	0.17	0.74 (0.37, 1.45)	0.35
Furry pet in house	1.25 (0.73, 2.16)	0.40	0.94 (0.67, 1.33)	0.72	0.82 (0.53, 1.27)	0.35	0.83 (0.45, 1.53)	0.53
*Aspergillus fumigatus*	0.86 (0.21, 3.45)	0.82	1.02 (0.61, 1.71)	0.94	0.73 (0.28, 1.87)	0.49	1.51 (0.52, 4.33)	0.42
Bla g 1	0.76 (0.38, 1.53)	0.42	0.86 (0.39, 1.91)	0.70	0.98 (0.48, 1.98)	0.95	0.98 (0.38, 2.52)	0.97
Bla g 2	0.89 (0.23, 3.52)	0.86	1.29 (0.55, 3.01)	0.53	2.14 (0.82, 5.56)	0.11	1.24 (0.41, 3.78)	0.69
Can f 1	0.25 (0.03, 2.27)	0.20	0.59 (0.25, 1.38)	0.20	0.44 (0.08, 2.24)	0.30	0.27 (0.05, 1.41)	0.11
Der f 1	2.00 (0.62, 6.48)	0.23	1.90 (1.02, 3.51)	0.04	1.34 (0.73, 2.50)	0.33	1.31 (0.42, 4.10)	0.62
Der p 1	2.32 (0.79, 6.88)	0.12	0.67 (0.25, 1.80)	0.41	6.69 (2.45, 18.31)	0.00	1.30 (0.48, 3.52)	0.58
Fel d 1	1.30 (0.24, 7.09)	0.74	1.25 (0.34, 4.61)	0.72	0.65 (0.19, 2.20)	0.46	4.42 (1.34, 14.56)	0.01
Mus m 1	1.14 (0.40, 3.27)	0.80	1.72 (0.69, 4.28)	0.22	1.35 (0.72, 2.55)	0.32	1.08 (0.36, 3.22)	0.88
Rat n 1	0.90 (0.36, 2.22)	0.80	1.02 (0.61, 1.70)	0.93	0.50 (0.19, 1.27)	0.13	0.45 (0.12, 1.63)	0.21
Endotoxin	1.60 (0.76, 3.41)	0.20	1.36 (0.59, 3.16)	0.45	1.03 (0.32, 3.32)	0.96	3.09 (1.59, 5.99)	0.00

Abbreviations: OR, odds ratio; CI, confidence interval; Bla g, *Blattella germanica*; Can f, *Canis familiaris*; Der f, *Dermatophagoides farinae*; Der p, *Dermatophagoides pteronyssinus*; Fel d, *Felis domesticus*; Mus m, Mouse urinary protein; Rat n, *Rat urinary protein*. Odds ratios (OR) calculated using logistic regression analyses adjusting for age, gender, race/ethnicity, family income to poverty ratio, environmental tobacco smoke, BMI, pets’ avoidance or removal, and potential confounding dust allergens. OR estimated for an increase in exposure from lowest to highest quartile of dust allergens concentrations (*Aspergillus fumigatus*, Bla g1 and 2, Can f1, Fel d1, Der p1, Der f1, Mus m1, Rat n1, endotoxin). Significant results are underlined.

**Table 3 ijerph-18-04945-t003:** Odds ratios (OR) and 95% confidence intervals (CI) for associations of dust components with asthma and wheeze by atopic status, NHANES 2005–2006.

Allergens	Atopic Asthma Status				Atopic Wheezing Status			
Dust Allergens	Non-Atopic Asthma		Atopic Asthma		Non-Atopic Wheezing		Atopic Wheezing	
	OR (95% CI)	*p*	OR (95% CI)	*p*	OR (95% CI)	*p*	OR (95% CI)	*p*
Mildew or musty smell	2.40 (1.38, 4.18)	0.00	1.42 (0.87, 2.34)	0.15	1.26 (0.77, 2.06)	0.33	1.17 (0.85, 1.60)	0.32
Cockroaches in house	0.73 (0.41, 1.29)	0.25	0.40 (0.23, 0.71)	0.00	0.87 (0.44,1.73)	0.68	0.92 (0.58, 1.47)	0.71
Dog in house	0.92 (0.48, 1.77)	0.79	0.67 (0.41, 1.12)	0.12	0.80 (0.47, 1.36)	0.38	1.10 (0.78, 1.55)	0.56
Cat in house	0.92 (0.66, 1.30)	0.63	1.31 (0.89, 1.93)	0.16	1.19 (0.94, 1.50)	0.13	1.08 (0.79, 1.48)	0.61
Furry pet in house	1.19 (0.91, 1.56)	0.19	0.87 (0.69, 1.10)	0.23	0.89 (0.61, 1.31)	0.53	1.09 (0.75, 1.58)	0.62
*Aspergillus fumigatus*	0.96 (0.57, 1.61)	0.87	0.77 (0.35, 1.70)	0.49	1.06 (0.73, 1.52)	0.75	1.03 (0.53, 2.01)	0.93
Bla g 1	1.09 (0.55, 2.16)	0.80	1.30 (0.71, 2.39)	0.37	1.02 (0.66, 1.60	0.91	1.23 (0.79, 1.91)	0.34
Bla g 2	0.63 (0.31, 1.29)	0.19	0.78 (0.48, 1.26)	0.28	1.01 (0.67, 1.51)	0.97	0.88 (0.57, 1.34)	0.51
Can f 1	1.12 (0.29, 4.39)	0.86	3.04 (1.21, 7.61)	0.02	1.15 (0.56, 2.37)	0.69	2.47 (1.40, 4.35)	0.00
Der f 1	0.38 (0.17, 0.87)	0.02	0.81 (0.43, 1.52)	0.49	0.55 (0.29, 1.03)	0.06	0.96 (0.56, 1.65)	0.87
Der p 1	1.50 (0.88, 2.55)	0.13	1.23 (0.59, 2.60)	0.56	1.19 (0.86, 1.64)	0.29	1.19 (0.62, 2.27)	0.58
Fel d 1	4.22 (1.52, 11.74)	0.00	0.68 (0.33, 1.42)	0.28	1.31 (0.57, 2.99)	0.50	1.22 (0.50, 2.96)	0.65
Mus m 1	0.81 (0.40, 1.65)	0.54	0.93 (0.45, 1.94)	0.84	1.02 (0.56, 1.89)	0.93	0.88 (0.47, 1.64)	0.67
Rat n 1	1.32 (0.65, 2.69)	0.41	1.29 (0.87, 1.93)	0.19	1.03 (0.63, 1.68)	0.91	1.16 (0.84, 1.62)	0.34
Endotoxin	1.05 (0.61, 1.83)	0.84	1.94 (1.29, 2.92)	0.00	1.45 (0.94, 2.25)	0.09	1.12 (0.61, 2.06)	0.69

Abbreviations: OR, odds ratio; CI, confidence interval; Bla g, *Blattella germanica*; Can f, *Canis familiaris*; Der f, *Dermatophagoides farinae*; Der p, *Dermatophagoides pteronyssinus*; Fel d, *Felis domesticus*; Mus m, mouse urinary protein; Rat n, r*at urinary protein*. Odds ratios (OR) calculated using logistic regression analyses adjusting for age, gender, race/ethnicity, family income to poverty ratio, environmental tobacco smoke, BMI, pets’ avoidance or removal, and potential confounding dust allergens. OR estimated for an increase in exposure from lowest to highest quartile of dust allergens concentrations (*Aspergillus fumigatus*, Bla g1 and 2, Can f1, Fel d1, Der p1, Der f1, Mus m1, Rat n1, endotoxin). Significant results are underlined.

## Data Availability

Data may be found on the NHANES website, accessed 1 February 2021 (http://www.cdc.gov/nhc/nhanes.htm).

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
