# Peer review of "Exposures in the Indoor Environment and Prevalence of Allergic Conditions in the United States of America"

_ijerph, 2021, doi:10.3390/ijerph18094945_

Round 1

Reviewer 1 Report

This is an interesting study aiming to examine the association between the presence of mildew, cockroaches and pets in homes as well as household dust allergens, and the prevalence and/or severity of allergic diseases. It has some innovation in the american context, namely in terms of simultaneously analysing several household exposures and the most common atopic conditions as well as their severity in a large sample of individuals and their homes.

However, there are some points that should be further clarified by the authors:

M&M

1) The authors used a questionnaire on environmental exposures which also included questions on the presence of mildew, cockroaches and pets in the home. The sensitivity of this self-reported information may be higher for pets and mildew (although lower in this case), but it might be low in the case of cockraches. Since the authors also measured cockroach and pet allergen levels in the dust (Bla g 2, Can f 1, Fel d 1), what were the related environmental questions used for. In other words, when controlling for exposure, were self-reports used or antigen levels. Please clarify.

2) Collection of house dust (Pg 3; lines 110-112): the authors state that ”A 1-square yard surface on both bed and adjacent floor was independently vacuumed for two minutes.”, to collect dust samples. However, it is not clear whetherthis procedure was different between houses with and without carpets, and this may be relevant. In fact, carpets concentrate more dust than non carpeted floors; thus, various groups (e.g. Heinrich et al, 2003) have standardised approaches for vacuuming carpeted (dusting 1m2) and non-carpeted floors (dusting the whole room). Please clarify whether both carpeted and non carpeted floors were only vacuum cleaned in 1 square yard. Please also indicate what was the percentage of homes with uncarpeted floors, so that na eventual bias in dust collection can be gauged.

3) Dust samples were also analysed for a mouse and rat allergen levels (Mus m 1, and Rat n 1, respectively). However, the questionnaire apparently did not use related quetsions for self-report. Please indicate whether questions on mouse and rat exposures were used as well or not.

4) Dust samples were only tested for Aspergillus but not Alternaria or Cladosporium allergens. Please clarify why these additional and increasingly important fungal aeroallergens, also at an indoor context, were not tested, since this may have failed to determine exposure to relevant fungal sources. If these allergens were not analysed, this should also be added to limitations.

5) In addition, black moulds (eg Stachybotrys chartarum) were apparently not tested; since these fungi are an increasingly acknowledged allergenic sources and may associated with worse asthma control in some cases (eg. Segura-Medina et al, Respir Med 2019; 150: 74-80), please cleraly indicate whether these fungal allergen sources were teste dor not. If not, this should also be added to limitations.

6) No information is given regarding when, during the year, were dust samples collected. Please add this information and also comment, in Discussion, whether results may have been biased by the time of the year most samples were collected (for instance, had they mostly been collected in the Summer, possibly no associations or different associations would be observed); the paper would definitely improved if a further sub-analyses of whether timing of collection of dust throughout the year (for instance , according to seasons) had any effect on observed results.

7) In formation on aeroallergen-specific IgE levels are not adequately placed (page 3; lines 125-130). They were most likely measured in the serum of tested individuals; however this is not clearly stated and, in fact, this information is misplaced under “House dust” sub-heading. Please confirm and clearly state that these levels are serum levels and place this information under a different sub-heading.

Results

8) Data shown were obtained from patients with a broad age range (6-85; mean 37 years); since all outcomes and spIgE parameters may be different between children/adolescents, and adults, it would be interesting to have a better idea of age stratification of tested individuals; please add this information; i tis particularly importante to realise the relative weight of children in the sample.

9) Still in connection with this issue, results may have been different between children and adults; please clarify whether there were any diferences; this sub-analysis is relevant and will be useful for interpretation of results.

10) Please clarify, on each table, whether it envolves multivariable analyses and adjusted odds ratios (OR) or just univariate analyses and non-adjusted OR.

11) Q: When assessing asthma severity (Table 2), dust-associated Der f1 was positively associated with 1 ER/doctor visit for wheeze; Der p1 with 2 visits; and mildew and/or musty smell, Fel d1, and endotoxin with 3 visits in past 12 months; these results are interesting but they would be possibly stronger if dose responses were observed. Please clarify whether there was a correlation between the dust concentrations of each molecular aeroallergen and  number of ER visits.

12) Results observed for the various variables are interesting and useful since the authors endeavoured to have an integrated view of exposures and their association with asthma and rhinitis, and were also careful enough to adjust the data for dust allergen levels. However, overall, their results have basically a classical (but improved) approach which is similar to that of other studies and feed into some discrepant results observed across such different studies with a similar focus.  The study might have been significantly improved if clusters of exposure had been analysed, as was done in the study by Mendy A, at al. Environ Health 2020; 19: 35. Doi: 10.1186/s12940-020-00585-y. Please use this reference and comment on this type of approach which may more adequately assess real life exposures, which result in clusters of exposure (for instance lower income homes may aggregate with highr exposure to mouse, rat, cokcroach or fungi, etc).

Minor Comments:

1) Pg 3 - sentences in lines 132-136 need to be rephrased because there is no joining particle;

2) pg 8, line 328 - “do” should be replaced by “dog”

3) pg 8; lines 334-337 should appear as a single sentence, not interrupted by a full stop.

Reviewer 2 Report

This manuscript evaluated the association between environmental exposures and allergic outcomes in a large population. It is worthy, and interesting.

Major

-How many children and adults were enrolled in this study?

 How many were diagnosed as asthma, allergic rhinitis, and other atopic diseases?

 Descriptive data of enrolled subjects should be suggested as a table 1, or supplementary table.

-How other atopic disease were diagnosed? Authors described that they assessed the presence of atopic diseases by questionnaire. Does it mean that enrolled subjects who were medically diagnosed as asthma, or allergic rhinitis were grouped as presence of atopic disease? Similar to the description about definition of current asthma in line 140, description regarding definition other atopic diseases such as allergic rhinitis should be added in ‘Assessment of Atopic Disease’ section.

-Is there any difference in results between children and adults?

-As authors’ description, this study is based on questionnaire from enrolled subjects. The assessment of atopic disease is dependent on enrolled subjects’ answers, and lacking in objective measurement. It should be added as a limitation of study in Discussion section.

Minor

-All abbreviations should be fully explained at its first usage (For example, ER, Th2, TLR...)

Round 2

Reviewer 2 Report

Authors tried to improve their manuscript.